# Mural Serum Response Factor (SRF) Deficiency Provides Insights into Retinal Vascular Functionality and Development

**DOI:** 10.3390/ijms241612597

**Published:** 2023-08-09

**Authors:** Alexander Günter, Vithiyanjali Sothilingam, Michael M. Orlich, Alfred Nordheim, Mathias W. Seeliger, Regine Mühlfriedel

**Affiliations:** 1Division of Ocular Neurodegeneration, Institute for Ophthalmic Research, University of Tübingen, 72076 Tübingen, Germany; vithiya@web.de (V.S.); see@uni-tuebingen.de (M.W.S.); 2Rudbeck Laboratory, Department of Immunology, Genetics and Pathology, Uppsala University, 75185 Uppsala, Sweden; michael.orlich@igp.uu.se; 3Department of Molecular Biology, Interfaculty Institute of Cell Biology, University of Tübingen, 72076 Tübingen, Germany; alfred.nordheim@uni-tuebingen.de

**Keywords:** serum response factor, mural cells, retinal ischemia, murine disease models, vascular smooth-muscle cells, retinal imaging

## Abstract

Serum response factor (SRF) controls the expression of muscle contraction and motility genes in mural cells (MCs) of the vasculature. In the retina, MC-SRF is important for correct angiogenesis during development and the continuing maintenance of the vascular tone. The purpose of this study was to provide further insights into the effects of MC SRF deficiency on the vasculature and function of the mature retina in *Srf^iMCKO^* mice that carry a MC-specific deletion of *Srf*. Retinal morphology and vascular integrity were analyzed in vivo via scanning laser ophthalmoscopy (SLO), angiography, and optical coherence tomography (OCT). Retinal function was evaluated with full-field electroretinography (ERG). We found that retinal blood vessels of these mutants exhibited different degrees of morphological and functional alterations. With increasing severity, we found vascular bulging, the formation of arteriovenous (AV) anastomoses, and ultimately, a retinal detachment (RD). The associated irregular retinal blood pressure and flow distribution eventually induced hypoxia, indicated by a negative ERG waveform shape. Further, the high frequency of interocular differences in the phenotype of individual *Srf^iMCKO^* mice points to a secondary nature of these developments far downstream of the genetic defect and rather dependent on the local retinal context.

## 1. Introduction

The retina is a neurosensory ocular tissue that detects light stimuli and translates them into electrical signals as a first step of visual perception [1]. To cover the high energy demand and to obtain sufficient nutrients and oxygen, a supporting dense and well-organized vascular network is formed during development [2]. When mature, the retinal vasculature metabolically sustains the inner retina, while the avascular outer retina is supplied by the choroidal vasculature [3]. In mice, developing retinal vessels are guided by an embryonic astrocytic network template [4]. Human and mouse vasculature development progresses in a similar way, with primary vessels originating from the optic nerve head and spreading across the retinal surface. In full-term humans, vascular development is complete at birth, while in mice, the development of retinal vessels takes place postnatally, paralleled by a regression of the hyaloid vasculature. Thus, pathological effects of genetic mutations that affect the retinal vasculature may be assessed in respective murine models well after birth [5]. 

Blood flow in the retina starts from the central artery, which enters the eye through the optic disc and ramifies into smaller arteriole branches that supply blood to capillaries, where the exchange of nutrients, gases, and waste with the retina takes place [6]. Post-capillary venules drain the venous blood to the central retinal vein, that in turn leaves the eye through the optic nerve. Pre-capillary arterioles and post-capillary venules are connected through the capillary bed [7]. Blood vessels are generally composed of endothelial cells that form the inner vascular wall and surrounding mural cells (MCs) [8]. MCs are subdivided into pericytes and vascular smooth-muscle cells (vSMC) (Figure 1A). In capillaries and post-capillary venules, pericytes are the only abundant MC type, while in larger-caliber vessels such as arteries, veins, and arterioles, vSMCs are majorly found [9,10,11]. In the mouse whole-mount retina, MCs are widely expressed, together with endothelial cells (Figure 1B). Functionally, pericytes are important for the vascular stabilization of growing blood vessels and for the establishment of the blood–retina barrier (BRB), while vSMCs are contractile elements that create the vascular tone, which is essential for the management of blood flow and the regulation of blood pressure [12,13].

In the cardiovascular system, MC dysfunction was shown to lead to atherosclerosis and aortic stiffness in hypertension [14]. In particular, aortic stiffness was found to be modulated by serum response factor (SRF), an important mediator of vSMCs’ mechanical properties [14]. SRF is a widely expressed transcription factor that has many roles, including muscle differentiation, cellular growth, and motility [15,16]. In the brain, it plays a central role in cerebral blood flow dysregulation in Alzheimer’s disease [17], while in the retina, impaired SRF levels of MCs were shown to be involved in the pathogenesis of ischemic retinopathy, leading to retinal hypoxia [18]. Many vascular diseases of the retina are connected to hypoxia, which plays a key role in the onset and progression of retinopathy of prematurity (ROP), age-related macular degeneration (AMD), and diabetic retinopathy [19], all major causes for acquired blindness worldwide. Further, neovascular forms of ischemic vascular diseases were also shown to ultimately lead to retinal detachment (RD) [20,21].

A general genetic deletion of *Srf* in mice is lethal, as it has an essential role in mesoderm formation [22]. Even a tissue-specific ablation of *Srf* in smooth-muscle cells resulted in lethal intestinal obstruction [23,24]. Recently, a non-lethal, MC-specific *Srf* knockout mouse model (*Srf^iMCKO^*) was introduced, in which changes to retinal vascular development were apparent. In particular, SRF was found to be critical for the regulation of the vascular tone in retinal vessels via the expression of contractile genes in vSMCs [18]. While this study focused on the effect of SRF during early postnatal vascular development, the aim of this work was to further investigate the implications of the lack of SRF on the maturing mouse in order to provide insights into factors that determine ischemic retinopathies.

For this purpose, retinal morphology and vascular integrity were investigated in adult *Srf^iMCKO^* mice via in vivo imaging and electroretinography (ERG). We observed that different mutant mice present vascular defects in distinct degrees of severity, including arteriovenous (AV) anastomoses that facilitated AV shunting in the retinal periphery. Many of these defects are associated with hypoxia, and thus a typical negative response waveform shape was present in the ERG examination of such eyes. Furthermore, major differences in the phenotype between the two eyes of the same *Srf^iMCKO^* mouse were frequently observed. Generally, a retinal degeneration with age was also found, featuring a progressive functional impairment over time. Overall, these data identify MC-SRF as a modulator of mechanical changes in the retinal vasculature with a marked downstream effect on the dependent structures. This leads to abnormal pressure increases as well as hypoxia in respective tissues, and eventually to a degeneration of the retina over time. Thus, the *Srf^iMCKO^* mouse is a suitable model to study the causative pathomechanisms of ischemic developmental vascular defects.

## 2. Results

### 2.1. In Vivo Imaging Reveals Vascular Defects of the Srf^iMCKO^ Mouse in Different Degrees of Severity 

For the analysis of the effect of a MC SRF deficiency on the retinal vasculature, retinal imaging was performed in mutant *Srf^iMCKO^* and control mice at four weeks of age. Native scanning laser ophthalmoscopy (SLO) imaging (Figure 2D,G,J) and indocyanine green (ICG) angiography (Figure 2E,H,K) showed different degrees of a bulging dilation of large surface vessels in *Srf^iMCKO^* mutant animals, not present in age-matched controls (Figure 2A–C). Furthermore, an apparent bulging of the microvasculature was generally found in the outer and the inner plexiform layers (OPL/IPL, respectively, Figure 2F,I,L). Abnormalities were graded as mild, intermediate, or severe based on in vivo morphology. In the mild condition (*n* = 5 eyes), there was only a slight thickening of large surface vessels in the fundus image (Figure 2D,E) and OPL/IPL microvasculature in the optical coherence tomography (OCT) (Figure 2F). The intermediate condition (*n* = 4 eyes) showed a more enhanced phenotype of the same kind (i.e., larger vessel diameters in Figure 2G–I, respectively). In severe cases (*n* = 5 eyes), an extreme bulging of the large surface vessels was found (Figure 2J,K), together with an extensively increased diameter of the downstream vasculature, so that affected vessels in the IPL appeared to extend into the inner nuclear layer (INL) (Figure 2L). At 8 weeks of age, the remaining *Srf^iMCKO^* eyes were graded as either intermediate (*n* = 3 eyes) or severe (*n* = 3 eyes).

### 2.2. Vascular Defects in the Srf^iMCKO^ Mouse Correlate with Retinal Function

Any potential functional consequences of the observed vascular alterations on the retina were assessed with ERG in *Srf^iMCKO^* in comparison to control mice at four weeks of age. The pathological change in the scotopic ERG responses corresponded well to the morphology-based classification into mild, intermediate, and severe degrees in the in vivo imaging data (Figure 3A). In particular, the shape of the ERG waveforms helped to determine the presence of retinal hypoxia based on differential changes in the negative-going a-wave and the positive-going b-wave. While the a-wave is, at least initially, driven mainly by photoreceptor outer segments, the b-wave primarily reflects ON bipolar cell activity [25]. Further, the photoreceptor layer is almost entirely supplied via choroid and retinal pigment epithelium (RPE), whereas the inner retina receives its vascular supply via the intraocular vessels. A primary defect in the retinal vasculature, as seen in the *Srf^iMCKO^* mutants, thus leads to a predominant reduction of the b-waves, and the overall degree of such a specific retinal hypoxia may be quantified via the b/a-wave amplitude ratio. In mild cases, this ratio did not show a reduction (overall) when compared to the control condition (Figure 3A,B). However, in mice with an intermediate degree of severity, a significant reduction in the b/a-wave ratio was observed as a sign of manifested hypoxia, and in severe cases, the ratio even fell below the value of one (Figure 3A,B). Such a so-called “negative” ERG waveform, i.e., where the b-wave is smaller than the a-wave [26], is indicative of a marked general retinal hypoxia under these conditions.

A common developmental feature found in a number of rodent models with retinal vascular pathology, such as ROP, Norrie disease, and angiopoietin-2 [27,28,29], is a retarded radial vessel outgrowth from the optic disc area during the formation of the superficial vascular plexus. The subsequent peripheral lack of capillary supply certainly also contributes to retinal hypoxia and, therefore, negative-type ERG data. In contrast, a specific finding in *Srf^iMCKO^* mice featuring a severe phenotype were peripheral retinal regions with direct, apparently premature connections between arteries and veins (‘AV shunts’), accompanied by a reduced microvascular network. Such alterations were not seen in control mice (Figure 3C,D) nor in the abovementioned rodent models [27,28,29]. The presence of these AV malformations suggests that the impaired muscular wall contraction ability, particularly on the arterial side, was compensated for by an increase in the vessel diameter at the cost of a highly increased volume downstream. AV shunts presumably form to reduce the volume load, but on the other hand, lead to a lack of capillary perfusion due to an ‘unused’ fraction of the blood flow through shunts, leading to hypoxia and malnourishment of the retinal cells in dependent regions.

### 2.3. Functional and Morphological Differences in Severity were Regularly Observed between the Two Eyes of the Same Srf^iMCKO^ Mouse

Over the course of the assessment of in vivo imaging and ERG responses in *Srf^iMCKO^* mice, we encountered functional and morphological differences in severity between the right and left eyes from the same individual animals on a regular basis. This was the case for 70% of mutants at postnatal week (PW) 4 and 100% at PW 8. In the example shown in Figure 4, major differences in the ERG between both eyes from the same *Srf^iMCKO^* mouse at PW 4 are apparent. Specifically, the condition in the right eye was classified as severe based on the presence of a negative scotopic ERG and lower b-wave amplitudes in both scotopic and photopic conditions when compared to control mice (Figure 4A,B). In contrast, b-wave shapes and amplitudes in the left eye of the same animal were similar to the control data (Figure 4A,B). A similar side difference was present in the in vivo morphology (SLO, angiography, and OCT imaging). The phenotype in the right eye was also in agreement with a severe condition with major alterations in large surface vessels, as well as the smaller vasculature in both the OPL and the IPL/INL (Figure 4C–E), similar to what was described in Figure 2J–L. The vessels in the microvasculature even became so large that their reflection became visible in the native SLO as white dots (Figure 4C). Conversely, the morphology in the left eye resembled a mild phenotype, with rather limited alterations to large retinal vessels and the microvasculature (Figure 4F–H), comparable to what was described in Figure 2D–F. 

The quite common occurrence of an interocular difference in morphological and functional data indicates that these are rather secondary developments, and the actual phenotype of individual *Srf^iMCKO^* mice is mainly dependent on the local retinal context.

### 2.4. The Phenotypic Spectrum in Srf^iMCKO^ Mice Includes a Retinal Detachment (RD)

The morphological and functional alterations due to changes in the retinal vasculature may lead to further complications in later adulthood. As an example, a partial detachment of the retina was occasionally observed in *Srf^iMCKO^* mutants at PW 8 via in vivo imaging (Figure 5A–C). Specifically, two out of five mice at PW 8 showed signs of RD, one of which had progressed to a full detachment. Due to the design of the multidisciplinary study and the maximum age at which we were able to test the animals, this was a rather rare endpoint, but it was definitely linked to the vascular disease as spontaneous detachments are extremely uncommon in mice. The data revealed the presence of an undetached portion of the retina in the upper (dorsal) region, while a large part of the retina was detached and, due to the small vitreous cavity in mice, was mostly in contact with the back of the lens, as evident in the OCT scan (Figure 5C). ICG angiography disclosed that the vasculature of the detached portion was still well-perfused (Figure 5B). 

Functionally, the scotopic ERGs recorded from the same eye were practically flat up to the highest flash intensities used (Figure 5D), rendering only minimal b-waves from 300 mcd*s/m^2^ to 30 cd*s/m^2^, in comparison to the normal control data (Figure 5E). However, it should be noted that the range of causes for this result, in addition to the more direct mutation-related damage, may include both technical reasons such as incomplete light projection to the retina and changes of the orientation of the electric field, as well as physiologic impairment, e.g., due to detachment from the RPE and reduced perfusion due to kinked vessels. 

## 3. Discussion

In this study, we further investigated the role of the MC-SRF transcription factor on the vasculature and function of the mature retina in *Srf^iMCKO^* mice that carry a MC-specific deletion of *Srf*. 

We found that impaired retinal blood vessels in these mutants led to different degrees of morphological and functional alterations. Vascular bulging, the formation of AV anastomoses, and ultimately, a RD, were associated with increasing severity. Retinal hypoxia was apparently correlated with the degree of irregular pressure and blood flow, indicated by ERG changes including amplitude reductions and alterations in the b/a-wave ratio up to a negative ERG waveform shape. We believe that these features may be primarily explained by alterations of the mechanical characteristics of vSMCs, together with the lack of vascular supply in the periphery due to an incomplete radial vascular development.

The ERG is a very good biomarker for retinal hypoxia, and a negative waveform shape has been described in many different rodent models featuring vascular defects, including ROP, Ang2^−/−^, and Norrie disease [27,28,29]. In disorders with a reduction of vascular capacity, the ERG a-wave is usually rather well-preserved, because photoreceptors are supplied by the extremely well-perfused choroidal vasculature. In contrast, the b-wave is much more sensitive to disease severity, as the inner retinal support has limited reserves to compensate for impairments of the retinal vasculature. Consequently, the gradual relative reduction of the b-wave with phenotype severity in our study up to a negative ERG in severe cases identifies these results as the functional consequences of a primarily retinal ischemia. In the mild condition, the b/a-wave ratio was not significantly reduced in comparison to the control condition, which indicates that compensatory mechanisms such as the slight thickening of large retinal vessels were widely efficient to compensate for the effects caused by the loss of the vascular tone. These mechanisms apparently fail in intermediate and severe degrees of the vascular alterations observed in the *Srf^iMCKO^* mouse.

Moreover, SRF is known to be an essential regulator of the contractile machinery of muscle cells [30]. In the vSMCs of *Srf^iMCKO^* mice, a deficit of the contractile machinery leads to a reduced ability to regulate the vascular tone, particularly on the arterial side. Therefore, blood flow cannot be well-controlled as the pressure is initially increased downstream, which the system attempts to compensate for with larger vessel diameters. The inherent behavior of vSMCs to constrict and dilate in response to changes in blood pressure is called the myogenic response, which is mainly performed by arterioles (Figure 1A) [31]. The increased blood flow without the presence of the myogenic response to pressure in upstream arterioles was, in our opinion, a key factor of the phenotype in the dependent vasculature, so insights from the *Srf^iMCKO^* mouse model may help to understand related pathology. For example, a defective myogenic response was found in diabetic retinopathy patients, leading to an acceleration of disease onset, and the associated dilation of the blood vessels was linked to vessel leakage and hemorrhages [32,33].

The most extreme phenotype in the *Srf^iMCKO^* mice was characterized by a large RD. In retinal diseases with ischemic characteristics, a detachment of the retina may be caused by different mechanisms. A tractional detachment commonly follows the formation of fibrovascular scar tissue in the vitreous that pulls the neurosensory retina from the RPE. In contrast, a major cause of an exudative detachment is the accumulation of fluid in the subretinal space [20,34], which has, e.g., been associated with an impaired function of pericytes, leading to the disruption of BRB integrity [20]. In the *Srf^iMCKO^* mouse, it appears plausible that the RD is exudative in nature. We reason that the volume increase, either particularly on the venous side directly, or due to a thinning of vascular walls associated with increased ultrafiltration, are possible causes. Alternatively, vascular malformations such as the AV anastomoses observed may eventually establish a tractional detachment. 

In addition, the commonly observed vascular and functional interocular differences in the same *Srf^iMCKO^* mice indicate the importance of the local retinal context for the development of the ultimate phenotype in this disorder. The original genetic defect, causing a lack of SRF expression in MCs, initially led to a loss of contractile flow control of the vSMCs, primarily on the arterial side. Subsequently, secondary changes were observed that were not directly associated with the gene mutation itself, but rather due to a dysfunction of primarily affected tissues. These changes include downstream vascular bulging, the formation of AV shunts, and ultimately, a partial RD. In addition to the rather mechanical effects on blood flow, the sensitivity of the retina to hypoxia may, e.g., via the reactive expression of vasoactive factors such as VEGF, further promote aberrant vascular developments, and these may also be considered secondary, or even tertiary (when due to the malformations), in nature. As the local retinal context varies to some degree in each individual eye, this may markedly influence the phenotype, and the less direct the effect of the original *Srf* mutation, the more epigenetic factors will play a role [35,36]. The post-transcriptional microRNA regulation of genes related to smooth-muscle cell proliferation and the response to hypoxia were shown to be significantly different between the right and left eyes in mice. This variance in eye laterality could be a regulatory factor that influences the interocular differences observed in the *Srf^iMCKO^* mice, since SRF is a transcription factor [37]. Another key element in this regard may be the high natural variability of the vasculature in mice, as the number of large arteries and veins varies between four and eight arterioles and venules each [38], and in particular, the number already impacts the diameter of the large veins, due to the presumably similar total blood volume per eye to be carried.

## 4. Materials and Methods

### 4.1. Animals

Housing of the animals was in agreement with the national guidelines for research animal housing (GV-SOLAS, Mainz, Germany). Mice were housed in an alternating 12 h light and dark cycle environment with free access to food and water, a relative humidity of 60%, and a temperature of 21 °C. They were held in type 2 long polycarbonate or in individually ventilated cages in groups of 2–5 animals per cage. New litters were weaned at PW 3 and separated into new cages. All animal experiments and procedures performed in this study adhered to the ARVO statement for the Use of Animals in Ophthalmic and Vision Research and were approved by the local animal ethics committee in accordance with the German Animal Welfare Act (permissions IM05/18G and IM02/19G by the regional authorities (Regierungspräsidium Tübingen, Germany). 

Transgenic *Pdgfrb (BAC)-CreERT2* [39] mice were bred with *Srf-flex1* mice [40] that carried a floxed exon 1 (flex1) of the *Srf* gene in order to obtain *Pdgfrb-CreERT2::Srf-flex1* mice on a C57BL/6 genetic background. A more detailed description of the generation of these mutants was previously published [18]. The conditional MC-specific deletion of *Srf* (referred to as *Srf^iMCKO^*) was temporally controlled with the injection of tamoxifen at postnatal days 1 to 3. The quantitative polymerase chain reaction (qPCR) of fluorescence-activated cell sorting (FACS) of MCs in *Srf^iMCKO^* mice showed an over 99% reduction in the mRNA expression of *Srf* [18]. For comparison, age-matched *Pdgfrb-CreERT2::Srf-flex1* mice without a tamoxifen injection were used as controls. For the experiments, 14 *Srf^iMCKO^* mice (9 males and 5 females) were used and compared with 10 control mice (6 males and 4 females). All animals were aged PW 4 or PW 8.

### 4.2. Electroretinography (ERG)

Mice were dark-adapted overnight. Subsequently, they were anesthetized with a subcutaneous injection of ketamine (66.7 mg/kg of bodyweight) mixed with xylazine (11.7 mg/kg of bodyweight), diluted in 0.9% NaCl saline, and they had their pupils dilated with tropicamide drops (Pharma Stulln, Stulln, Germany). All handling procedures were performed under dim-red light and recordings were conducted under dark conditions. The animals were positioned on a temperature-controlled mat and gold-wire ring electrodes, moisturized with methylcellulose (OmniVision GmbH, Puchheim, Germany), contacted the surface of both corneas for binocular recordings. Two short stainless-steel needle electrodes (Sei Emg s.r.l., Cittadella, Italy) were used as the reference and ground electrodes. Full-field ERG recordings were performed with the Espion E^3^ console featuring a computer, a 32-bit amplifier, and a Ganzfeld Bowl (Diagnosys, LLC, Lowell, MA, USA). Upper and lower filter frequency cutoffs of the amplifier were set to 0.3 Hz and 300 Hz, respectively. Initially, a dark-adapted single-flash ERG series, ranging from 1 mcd*s/m^2^ to 30 cd*s/m^2^, was recorded. Following a light adaptation of 10 min with a rod-saturating 30 cd*s/m^2^ background, a light-adapted single-flash ERG series, ranging from 10 mcd*s/m^2^ to 30 cd*s/m^2^, was recorded on the same background. Each recording step was averaged 10–15 times, with inter-stimulus intervals of 2 s (1–300 mcd*s/m^2^), 5 s (1–3 cd*s/m^2^), or 10 s (10–30 cd*s/m^2^). B-wave amplitudes were measured from the through of the a-wave to the peak of the b-wave.

### 4.3. Scanning Laser Ophthalmoscopy (SLO) and Optical Coherence Tomography (OCT)

In vivo imaging was performed as previously described [41,42,43]. Animals were examined immediately after the ERG session under the same anesthesia. A custom-made 100 dpt contact lens was placed on the cornea after application of hydroxypropyl methylcellulose to avoid dehydration. SLO imaging was performed together with OCT on the Spectralis™ HRA + OCT device using the proprietary software package Eye Explorer version 5.3.3.0 (Heidelberg Engineering, Heidelberg, Germany). The OCT section features a super-luminescent diode at 870 nm as a low-coherence light source. Each two-dimensional B-Scan recorded at a 30° field of view contains up to 1536 A-Scans, which are acquired at a speed of 40,000 scans per second. For fluorescence angiography, 75 mg/kg bodyweight of fluorescein (FLA) was subcutaneously injected, and images were generated with a blue (488 nm) stimulating laser and a barrier filter at 500 nm. For ICG angiography, 50 mg/kg bodyweight of ICG was subcutaneously injected, and images were generated with an infrared (795 nm) stimulating laser and a barrier filter at 800 nm. Data were exported as 8-bit grey-scale image files, and the scale bars were calibrated based on OCT and SLO retina images with intraocular-injected beads of a defined diameter [44].

### 4.4. Data and Statistical Analysis

The degree of severity of the retinal morphology was based on SLO, angiographic, and OCT images, regarding (a) the appearance of surface vessels (thickness, bulging dilation), (b) the degree of pathologically widened smaller vasculature in both the OPL and the IPL/INL, and (c) the formation of AV anastomoses. Mild: no major abnormalities (a)–(c), intermediate: major changes in one or more but not all aspects, and severe: significant changes in all aspects (a)–(c). The comparisons of the b/a-wave amplitude ratios between *Srf^iMCKO^* mutants and controls were performed with a one-tailed, two-sample Student’s *t*-test, assuming unequal variances. Values of *p* < 0.05 were considered statistically significant and labeled with an asterisk (*) in the graphs. Values of *p* < 0.01 were marked with two (**) asterisks to indicate a higher degree of statistical significance. Box-and-whisker plots and statistical analyses were performed with Microsoft Excel (Microsoft Corp., Albuquerque, NM, USA). The figures were prepared using the CorelDRAW X5 software (Corel corporation, Ottawa, ON, Canada). The diagram was created with BioRender.com.

## 5. Conclusions

This work highlighted the importance of MC-SRF for the development of the retinal vasculature and its integrity in the mature *Srf^iMCKO^* murine model. We found that MC-SRF had a relatively immediate impact on the formation of a correct vascular topography and the maintenance of muscular blood flow regulation. Secondary and tertiary key features of the retinal disease outcome in *Srf* mutants were downstream blood vessel dilatation, retinal hypoxia, and in severe cases, RD. These key features were widely shaped by the local environment, leading to an increased variability in the progression and severity of the phenotype, and often even to substantial interocular differences in the same animal. Further, this study illustrated the importance of a combined assessment of the in vivo function and morphology of the retina to obtain a comprehensive understanding of disease effects in mutant models of retinal vascular disorders. 

## Figures and Tables

**Figure 1 ijms-24-12597-f001:**
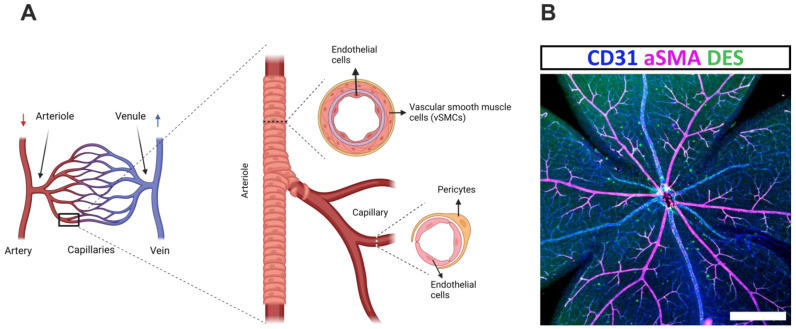
Mural cell (MC) coverage of retinal blood vessels. (**A**) The retinal arteries transport oxygen-rich blood (red arrow) to arterioles and capillaries, where the exchange of gases, nutrients, and waste with the retina takes place. Oxygen-poor blood returns through venules and veins to the systemic circulation (blue arrow). An enlargement of the outlined area shows arterioles and ramifying capillaries, generally constituted of endothelial cells and MCs. Vascular smooth-muscle cells (vSMC) are present in arterioles and larger-caliber vessels, while pericytes are found in capillaries and smaller-caliber vessels. (**B**) Confocal image of an adult mouse retina stained for the endothelial-specific marker CD31, the smooth-muscle cell-specific marker, alpha-smooth-muscle actin (aSMA), and the MC-specific marker Desmin (DES). Scale bar = 500 µm.

**Figure 2 ijms-24-12597-f002:**
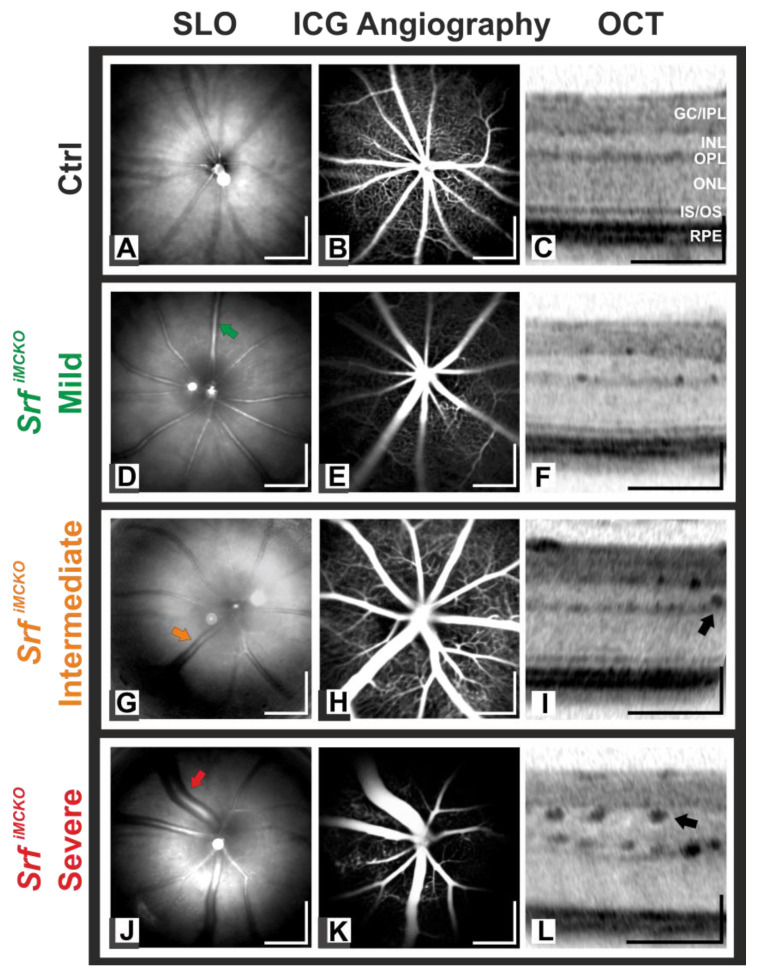
In vivo imaging of the retinal vasculature and retinal layer morphology of representative *Srf^iMCKO^* and control mice at four weeks of age, with scanning laser ophthalmoscopy (SLO) and optical coherence tomography (OCT). Native SLO (**D**,**G**,**J**) and indocyanine green (ICG) (**E**,**H**,**K**) images revealed different degrees of vascular bulging of the large blood vessels in *Srf^iMCKO^* mice (colored arrows) in comparison to control mice (**A**,**B**). In the OCT images (**F**,**I**,**L**), corresponding variations in the severity of the microvascular alterations (black arrows) were seen when compared to the control (**C**). GC/IPL: ganglion cell/inner plexiform layer, INL: inner nuclear layer, OPL: outer plexiform layer, ONL: outer nuclear layer, IS/OS: inner/outer segment, RPE: retinal pigmented epithelium. Scale bar = 500 µm (SLO and angiography images) and 150 µm (OCT images).

**Figure 3 ijms-24-12597-f003:**
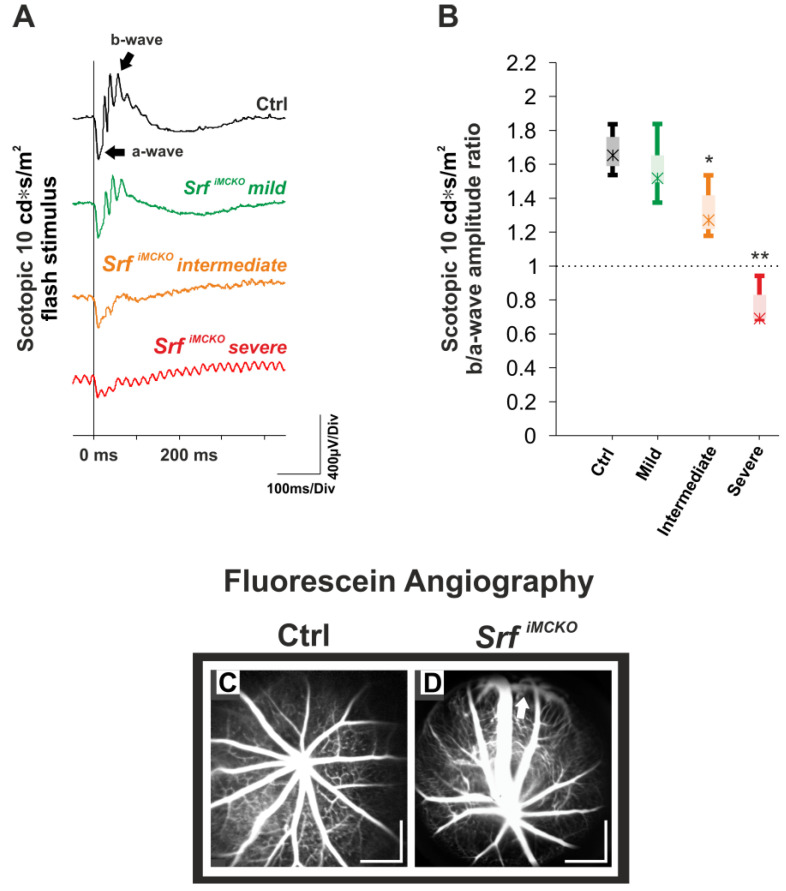
Retinal function in *Srf^iMCKO^* mutants and controls at four weeks of age. (**A**) Representative scotopic 10 cd*s/m^2^ ERG traces show a reduction of the b-wave amplitude with an increase of severity. (**B**) Box-and-whisker plots to illustrate the reduction in ERG b/a-wave amplitude ratios. In mild cases (*n* = 10 eyes), the overall ratio already showed a trend, but was not significantly reduced. In intermediate cases (*n* = 3 eyes), the b/a-wave ratio was lower than in control animals (*n* = 13 eyes). In severe cases (*n* = 3 eyes), the b/a-wave amplitude ratio was even below 1 (‘negative ERG’). (**C,D**) Representative in vivo fluorescein (FLA) angiography images highlight arteriovenous (AV) anastomoses (arrow) in severely affected *Srf^iMCKO^* mice (**D**), which were not observed in control mice (**C**). Statistically significant differences are indicated with asterisks (* *p* < 0.05 and ** *p* < 0.01). Boxes: 25–75% quantile range, whiskers: 5% and 95% quantiles, asterisks: median. Scale bar = 500 µm.

**Figure 4 ijms-24-12597-f004:**
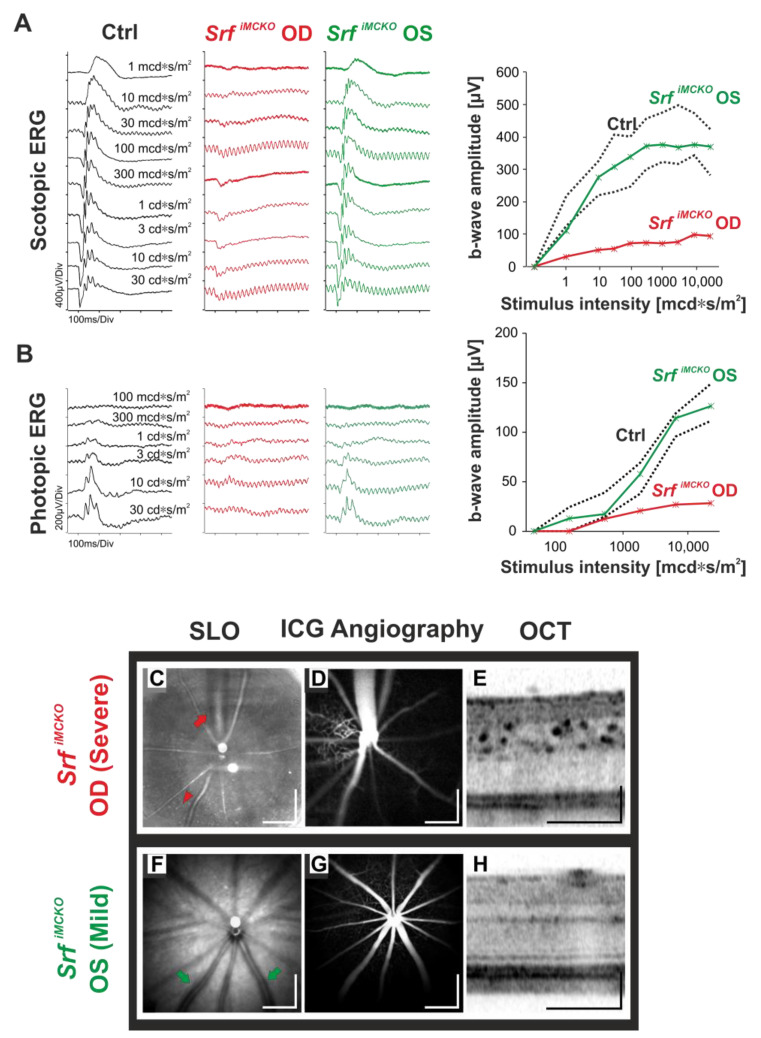
Functional and morphological interocular differences in a *Srf^iMCKO^* mutant at four weeks of age. (**A**,**B**) Representative scotopic and photopic single-flash-intensity series in the right (OD) and left (OS) eyes of the same *Srf^iMCKO^* mouse, in comparison to a normal control, plus a quantification of scotopic and photopic b-wave amplitudes to illustrate the side difference. Red and green colors were used to designate the data of OD and OS, respectively. For comparison, the amplitude data range of control mice (*n* = 8 eyes) is indicated by black dotted lines that mark the 5% (lower black trace) and 95% (upper black trace) quantiles. Native SLO (**C**,**F**), ICG angiographic (**D**,**G**), and OCT (**E**,**H**) images from the same mouse demonstrate a corresponding morphological side difference with respect to the vascular phenotype, both in large (colored arrows) and small (red arrowhead) vessels. Scale bar = 500 µm (SLO and angiography images) and 150 µm (OCT images).

**Figure 5 ijms-24-12597-f005:**
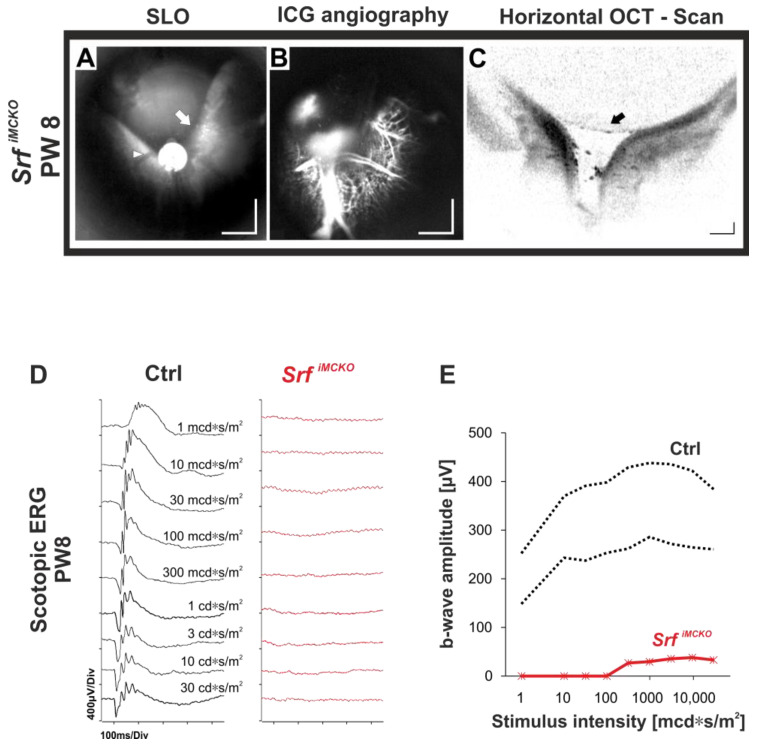
Example of in vivo imaging and ERG in a *Srf^iMCKO^* mouse featuring a partial retinal detachment (RD) at eight weeks of age. The partial detachment (white arrow) is well-visible in the native SLO (**A**). Retinal blood vessels may be seen on the surface of the detached retina (arrowhead). In ICG angiography (**B**), both large and small vasculature in the detached retina became visible. A horizontal OCT scan through the optic nerve head (**C**) disclosed the position of both detached and undetached portions of the retina. Further, the back of the lens is visible (black arrow). (**D**) Scotopic single-flash-intensity series from a *Srf^iMCKO^* animal eye (red curves) with RD and a control mouse (black curves). (**E**) Quantification of the scotopic b-wave amplitudes showed a severe reduction of the ERG responses after the detachment of the retina (red line). For comparison, the amplitude data range of control mice (*n* = 8 eyes) is indicated by black dotted lines that mark the 5% (lower trace) and 95% (upper trace) quantiles. Scale bar = 500 µm (SLO and angiography image) and 150 µm (OCT image).

## Data Availability

Not applicable.

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
