# Peer review of "Mural Serum Response Factor (SRF) Deficiency Provides Insights into Retinal Vascular Functionality and Development"

_ijms, 2023, doi:10.3390/ijms241612597_

Round 1

Reviewer 1 Report

The authors provide an interesting manuscript on the role of Mural serum response factor deficiency in retinal vascular development and function. Although the current manuscript is partly an extension of Orlich et al., 2022 paper, the manuscript is clear and they do a nice job in presenting data. The authors should add additional details. Below are my comments.

# in the current paper author analyzed for 4 and 8 weeks, what happens to the phenotype in the older animals?

# Can authors provide the supplementary image for mRNA expression of srf in KO compared to the control?

# it would be beneficial to the readers if authors can provide the mural cell coverage image from the retina.

Reviewer 2 Report

Summary:

In this study authors have used previously established mouse model (SrfiMCKO) that carry a Mular Cell-specific deletion of serum response factor (SRF) to characterize effect of SRF deficiency on the retinal vasculature and function in adolescent mice. They have utilized scanning laser ophthalmoscopy (SLO), angiography, optical coherence tomography (OCT) and full-field electroretinography (ERG) to evaluate retinal morphology, vasculature, and function. Both male and female mice were used in experiments. They found variable degree of differences in both retinal morphology and vasculature. Also, they observed impaired retinal function in accordance with severity of such alterations. Other possible causes of impaired retinal function in the SrfiMCKO mice are also acknowledged.

Overall, this is a well written draft. The study has been designed and conducted aptly, results are clearly explained and discussed. I recommend it to accept for publishing once the following minor concerns are fulfilled.

I only have following minor concerns that need to be address.

1.     Scale bars are missing in all pictures in figures 1, 2 and 3.

2.     In retina pictures colored arrow-head mark would make the respective phenotype more noticeable

3.     Is it possible to quantify of vessel diameter and compare the degree of abnormality to define “mild, intermediate and severe” phenotype in your study?

4.     Figure 2: labels C & D should be in similar format to that of A & B, e.g “C” positioned at top-left

5.     Figure 3: labels C & D should be in similar format to that of A & B, e.g “C” positioned at top-left

6.     Result 2.3: What percentage of the animal showed Functional and morphological differences in severity? (Does “regularly” meant to say 100% of the time?)

7.     Result 2.3: Were the Functional and morphological differences in severity at higher/lower frequency among male & female?

8.     Result 2.4: Was the phenotype retinal detachment observed/occurred 100% of the time?

Reviewer 3 Report

In this paper, the authors investigated the role of mural cell SRF deficiency on vasculature and function of the mature retina. They found different degrees of retinal blood vessels abnormal and functional alterations in adult SrfiMCKO mice. Overall, the results are interesting. However, there are several issues that should be addressed. 

1.    The authors found different degrees of retinal morphology abnormals. Then how to define the mild, intermediate, and severe degree? The authors should describe the grading system or grading standard in methods section.

2.    The Figure 1 only showed typical retinal morphology for each group. Then how many eyes were included in each degree? Did the number of eyes show any different between 4 and 8 weeks? How many eyes progressed to retinal detachment at 8 weeks?

3.    In Figure 2, panel C and D were fluorescein angiography images. While Figure1, 3 and 4 inculed ICG angiography images. Why did the authors choose two different angiography? Did the result show any different?

4.    The results showed that deletion of Srf in mural cell led to different degrees of retina morphological and functional alterations. Even in the same animal, the two eyes still could exhibit varying degrees of severity. These results indicated the existence of other regulatory factors that affect the development of retinal blood vessels and functions. The authors shoud discuss more about this.
